# Lightweight Spatio-Temporal Modeling via Temporally Shifted Distillation for Real-Time Accident Anticipation

**Patrik Patera, Yie-Tarng Chen, Wen-Hsien Fang**
National Taiwan University of Science and Technology
{d10902804,ytchen,whf}@mail.ntust.edu.tw

## Abstract

Anticipating traffic accidents in real time is critical for intelligent transportation systems, yet remains challenging under edge-device constraints. We propose a lightweight spatio-temporal framework that introduces a temporally shifted distillation strategy, enabling a student model to acquire predictive temporal dynamics from a frozen image-based teacher without requiring a video pre-trained teacher. The student combines a RepMixer spatial encoding with a RWKV-inspired recurrent module for efficient long-range temporal reasoning. To enhance robustness under partial observability, we design a masking memory strategy that leverages memory retention to reconstruct missing visual tokens, effectively simulating occlusions and future events. In addition, multi-modal vision-language supervision enriches semantic grounding. Our framework achieves state-of-the-art performance on multiple real-world dashcam benchmarks while sustaining real-time inference on resource-limited platforms such as the NVIDIA Jetson Orin Nano. Remarkably, it is 3-7× smaller than leading approaches yet delivers superior accuracy and earlier anticipation, underscoring its practicality for deployment in intelligent vehicles.

## 1 Introduction

Anticipating traffic accidents in real time involves assigning a confidence score to each video frame that reflects the likelihood of an imminent incident. This task is challenging due to the narrow temporal window for prediction and the rapidly changing dynamics of real-world driving environments. Accidents often occur suddenly and span only a brief segment of a video, making early detection difficult without compromising precision. Additionally, unpredictable driver behavior, occlusions, and visual clutter complicate the modeling of spatio-temporal cues in a reliable and timely manner.

Early approaches relied on RNN-based architectures with soft-attention mechanisms, such as DSA (Chan et al., 2016) and FA (Fatima et al., 2021), but lacked strong spatial reasoning capabilities. AdaLEA (Suzuki et al., 2018) improved early prediction supervision using a Quasi-RNN with adaptive penalties, yet remained sensitive to occlusions. More recent methods introduced graph-based modeling or reinforcement learning to enhance relational understanding and context-awareness

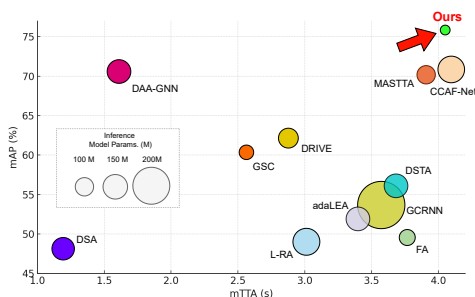

Figure 1: **Accuracy–anticipation trade-off**. The x-axis shows mean time-to-accident (mTTA, higher better) and the y-axis shows mean average precision (mAP). Bubble size corresponds to the number of model parameters. Our lightweight model (green) achieves superior early anticipation and accuracy while requiring significantly fewer parameters than larger baselines.

(Bao et al., 2020; Zeng et al., 2017; Bao et al., 2021). However, these approaches often rely on predefined graph structures, dense object-level annotations, or multi-stage pipelines involving detection and tracking, limiting robustness and practical deployment in unstructured, real-time settings. Other models such as DSTA (Karim et al., 2022) and

GSC (Wang et al., 2024) refined spatio-temporal modeling via region-of-interest selection and occlusion handling. More recently, CCAF-Net (Liu et al., 2025) fused RGB and depth features with a complementarity-aware attention mechanism, improving accuracy at the price of a large model and extra depth information. Many existing methods focus narrowly on either object-level semantics or global temporal cues, limiting their generalization to diverse driving scenarios. In addition, they often depend on heavy object-centric, multi-stage architectures – performing object detection with Faster R-CNN followed by feature extraction with VGG16 – which are computationally expensive and lack an end-to-end design.

To address these challenges, we propose a lightweight spatio-temporal distillation framework designed for real-time deployment in low-resource environments. Unlike conventional object-centric multi-stage pipelines, our approach is a compact, end-to-end model that operates directly on raw RGB frames. Traditional pipelines incur high latency and computational overhead, limiting their suitability for embedded systems. In contrast, our framework introduces a temporally shifted distillation strategy that lets a lightweight student learn temporal dynamics from a frozen image-only MobileCLIP (Vasu et al., 2024) teacher, eliminating the need for a temporally aware teacher.

At the core of our method, the student predicts future visual cues by aligning with temporally shifted teacher features. This supervision is applied exclusively within spatio-temporal modules, enabling temporal reasoning even though the teacher lacks temporal context. To further enhance representation under partial observability, the student is trained to reconstruct masked visual tokens using its recurrent hidden state, effectively simulating occlusion and reinforcing temporal abstraction.

Our student adopts a hybrid architecture that combines early RepMixer layers (Vasu et al., 2024) for efficient spatial encoding with a lightweight window-based spatio-temporal module adapted from the Receptance Weighted Key Value (RWKV) block (Peng et al., 2023). Our adaptation incorporates masking-aware recurrence, enabling robust long-range temporal modeling with linear attention complexity and memory retention. This design ensures real-time inference on embedded devices.

Training begins with a pre-training stage that combines temporally shifted distillation and contrastive learning on paired video–text data. The distillation transfers predictive temporal cues from a frozen image-only MobileCLIP (Vasu et al., 2024) teacher, while contrastive supervision aligns frame-level features with accident-related textual prompts to enrich semantic understanding and improve generalization. The model is then fine-tuned end-to-end on accident anticipation benchmarks, resulting in a compact, interpretable architecture that achieves state-of-the-art performance on real-world dashcam datasets and runs efficiently on platforms such as the NVIDIA Jetson Orin Nano, as shown in Figure 1.

**The major contributions of this paper are:**

- A **temporally shifted distillation framework** that enables spatio-temporal learning from a frozen image-based teacher, eliminating the need for a temporally aware teacher in video pre-training and making the approach suitable for small datasets and low-resource settings.
- A **lightweight hybrid student architecture** that integrates RepMixer spatial encoding with a recurrent temporal module, RWKV, providing efficient long-range video understanding with linear-time complexity.
- An adaptation of the RWKV block into a **window-based, mask-aware spatio-temporal module**, which integrates localized recurrence with the proposed masked memory strategy to achieve robust temporal modeling under occlusion and partial observability.
- **Real-time accident anticipation** with state-of-the-art performance on real-world benchmarks, running efficiently on the NVIDIA Jetson Orin Nano. Our model is 3–7× smaller than recent leading approaches, yet achieves high anticipation performance while remaining deployable on resource-constrained platforms.

## 2 RELATED WORK

### 2.1 ACCIDENT DETECTION

(Roy et al., 2022) used a Siamese network with LSTM and temporal attention for collision detection, while (Yu et al., 2024) employed a knowledge distillation-based framework to improve accident re-

gion identification and severity estimation. Unlike these approaches, which rely on bird's-eye view detection or distillation techniques, our approach employs pre-trained models paired with specialized adapters, providing a more efficient solution tailored for first-person driving contexts. (Fang et al., 2022) proposed a self-supervised consistency model for capturing spatio-temporal visual relationships, while (Zhou et al., 2022) introduced a neural network that clusters frames based on temporal features to detect accidents.

## 2.2 MULTI-MODAL LEARNING

(Wu et al., 2024) used a pre-trained CLIP model with learnable prompts and local-global modules for weakly-supervised video anomaly detection. (Singh & Mohan, 2019) trained denoising autoencoders on non-accident traffic videos, incorporating multi-modal inputs like frames and optical flow. (Wu et al., 2022) employed 3D CNNs to combine frames and depth data for spatio-temporal modeling in action recognition. In contrast, our approach focuses on video-text feature pairs to streamline real-time accident anticipation. (Huang et al., 2024) used a frozen CLIP model for image-to-video adaptation in action recognition. Unlike these methods, which focus on action recognition, our work extends this concept to anticipate traffic accidents.

## 2.3 RECURRENT TRANSFORMERS

Recent works have explored combining Transformer and recurrent network advantages. (Katharopoulos et al., 2020) proposed a linear attention formulation that reduces memory usage and supports recurrent inference. (Zhai et al., 2021) introduced an attention-free Transformer with linear complexity, validated on autoregressive and classification tasks. In video analysis, (Yang et al., 2022) presented a fully recurrent vision transformer with attention gating for long-clip training. VR-WKV (Duan et al., 2024) extends RWKV (Peng et al., 2023) to vision using quad-directional shifts and bidirectional global attention, though its bi-directionality limits real-time applicability

## 3 PROPOSED METHOD

We propose a lightweight spatio-temporal framework for real-time accident anticipation, designed for efficiency on embedded devices while maintaining predictive accuracy. The framework has three key components:

1. A **hybrid student architecture** combining efficient spatial encoding with recurrent temporal modeling.
2. A **window-based spatio-temporal block with masked recurrence** that integrates temporal and channel mixing.
3. A **temporally shifted distillation strategy** that enables predictive learning from a frozen image-based teacher without requiring large-scale video pre-training.

Together, these components allow the model to capture sudden accident dynamics while operating under strict latency and memory constraints.

## 3.1 HYBRID SPATIO-TEMPORAL STUDENT ARCHITECTURE

Purely spatial models lack temporal reasoning, while full spatio-temporal Transformers scale quadratically in both space and time $O(N^2T^2)$, making them impractical for real-time systems due to high computational and memory costs. To balance expressiveness and efficiency, we design a **hybrid student model**, illustrated in Figure 2, composed of three main components:

**Spatial Encoding (RepMixer).** Each frame is partitioned into non-overlapping patches and tokenized. Tokens are processed by RepMixer blocks adapted from MobileCLIP (Vasu et al., 2024), combining depthwise convolutions, normalization, residual connections, and lightweight MLP layers. This captures fine-grained traffic semantics (e.g., lanes, vehicles, pedestrians). Resolution is progressively reduced and channel depth increased, forming a compact hierarchy optimized for real-time inference. A frozen MobileCLIP text encoder provides multi-modal supervision through video–language alignment.

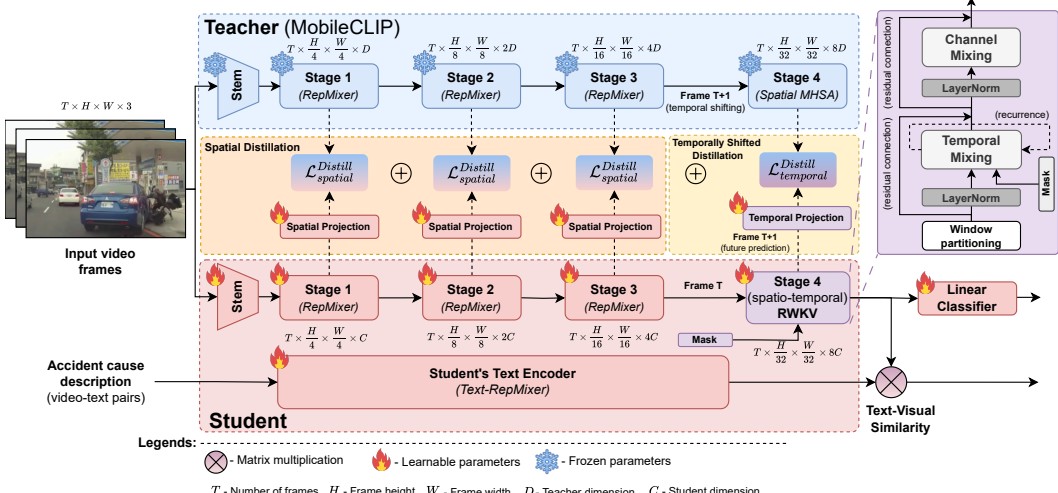

Figure 2: **Overview of our teacher–student framework**. The teacher is a frozen MobileCLIP model with four RepMixer stages (Stage 4 uses spatial-only MHSA). The student shares the same backbone but replaces Stage 4 with a spatio-temporal RWKV block for efficient temporal reasoning. Spatial distillation is applied at Stages 1–3, while temporal distillation aligns the student's current output at frame $T$ with the teacher's future features at frame $T+1$. Masked recurrence within the spatio-temporal RWKV block (right) simulates occlusion and strengthens memory retention.

**Temporal Encoding (Window-based Spatio-Temporal RWKV).** Instead of quadratic self-attention, we design a spatio-temporal RWKV block that maintains hidden states across frames with linear complexity. Processing occurs within local spatial windows, preserving spatial structure while propagating temporal memory. Learnable time-decay parameters enable long-range dependency modeling while supporting parallelized training.

**Masked Memory Strategy.** To improve robustness under occlusion and partial observability, a binary mask is introduced into the recurrence. During training, some tokens are blocked, forcing the model to rely on prior hidden states. This simulates real-world conditions (e.g., occluded pedestrians, motion blur) and strengthens predictive reasoning.

## 3.2 WINDOW-BASED SPATIO-TEMPORAL BLOCK

We introduce a window-based spatio-temporal recurrent block, inspired by RWKV (Peng et al., 2023), which achieves linear complexity by replacing attention with localized recurrence. Input features are first divided into $K$ non-overlapping windows of size $p_1 \times p_2$ and reshaped into sequences of shape $\mathbb{R}^{(B \cdot K) \times (T \cdot p_1 \cdot p_2) \times C}$ for temporal recurrence. The block comprises two complementary modules – Temporal Mixing and Channel Mixing – that together enable efficient modeling of temporal dependencies and spatial dynamics.

### 3.2.1 TEMPORAL MIXING

To model temporal dependencies efficiently, we apply a linear mixing of current $X_t$ and previous $X_{t-1}$ frame information:

$$R_t = \mathbf{W}_r(\mu_r X_t + (1 - \mu_r)X_{t-1}), \tag{1}$$

$$K_t = \mathbf{W}_k(\mu_k X_t + (1 - \mu_k)X_{t-1}), \tag{2}$$

$$V_t = \mathbf{W}_v(\mu_v X_t + (1 - \mu_v)X_{t-1}), \tag{3}$$

where $\mu_r, \mu_k, \mu_v \in \mathbb{R}^C$ are learnable mixing coefficients, and $\mathbf{W}_r, \mathbf{W}_k, \mathbf{W}_v \in \mathbb{R}^{C \times C}$ are projection matrices.

At each time step, hidden states accumulate past information with learnable decay factors of current visual tokens $k_t$ and $v_t$ and input mask $m_t$:

$$wkv_t = \frac{s_{t-1} + m_t \odot (e^{u+k_t} \odot v_t)}{d_{t-1} + m_t \odot e^{u+k_t}}, \tag{4}$$

$$s_t = m_t \odot \left(e^{-w} \odot s_{t-1} + e^{k_t} \odot v_t\right) + (1 - m_t) \odot s_{t-1}, \tag{5}$$

$$d_t = m_t \odot \left(e^{-w} \odot d_{t-1} + e^{k_t}\right) + (1 - m_t) \odot d_{t-1}, \tag{6}$$

where $w, u \in \mathbb{R}^{K \times C}$, with $K$ denoting the number of windows, are learnable time decay parameters and $\odot$ denotes element-wise multiplication. The recurrent output is gated by $R_t$ and projected:

$$rwkv_t = \mathbf{W}_o(\sigma(R_t) \odot wkv_t), \tag{7}$$

with $\mathbf{W}_o \in \mathbb{R}^{C \times C}$ a projection matrix and $\sigma$ a sigmoid gate function. This formulation provides long-range temporal expressiveness at linear cost.

### 3.2.2 CHANNEL MIXING

Channel mixing enriches intra-frame representations by modeling non-linear interactions across feature channels. For input $X_t$:

$$R'_t = \mathbf{W}'_r(\mu'_r X_t + (1 - \mu'_r) X_{t-1}), \tag{8}$$

$$K'_t = \mathbf{W}'_k(\mu'_k X_t + (1 - \mu'_k) X_{t-1}), \tag{9}$$

$$cmix_t = \sigma(R'_t) \odot \mathbf{W}'_o \left(\text{ReLU}(K'_t)^2\right), \tag{10}$$

where $\mu'_r, \mu'_k \in \mathbb{R}^C$ are learnable interpolation coefficients, and $\mathbf{W}'_r, \mathbf{W}'_k, \mathbf{W}'_o \in \mathbb{R}^{C \times C}$ are projection matrices. Interpolation smooths features across time, while the squared ReLU activation improves stability and expressiveness.

### 3.3 MASKED MEMORY STRATEGY

Driving scenes often exhibit partial observability: pedestrians hidden behind vehicles, motion blur, or poor illumination. To improve robustness under such conditions, we introduce a masking strategy into the Temporal Mixing module (Section 3.2.1).

At each time step, a binary mask $m_t \in \{0, 1\}$ determines whether the current visual tokens are incorporated into the $WKV_t$ computation and whether the hidden state is updated:

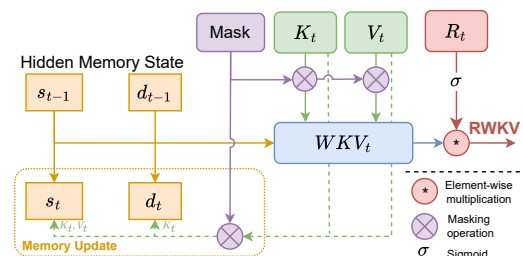

Figure 3: **Masking Strategy in Spatio-temporal RWKV.**

- If $m_t = 1$: the update and $WKV_t$ exploit the key–value pairs $(K_t, V_t)$.
- If $m_t = 0$: the update propagates only the prior memory $(s_{t-1}, d_{t-1})$.

This acts as a form of **memory dropout**: the model sometimes must "remember" rather than "see." Figure 3 illustrates how masked steps bypass the current frame while selectively updating hidden states with a binary mask, enabling conditional temporal propagation across spatio-temporal windows. This masking strategy is used only during pre-training; both fine-tuning and inference operate with the unmasked formulation. The masked memory update is applied once per forward pass in pre-training, without introducing extra stages or additional unrolled iterations. All operations – including masking – are implemented directly within our fused CUDA kernel, resulting in negligible computational overhead.

### 3.4 TEMPORALLY SHIFTED DISTILLATION

Large-scale video pre-training is costly and often impractical for rare events such as accidents. We propose a **temporally shifted distillation framework** that enables predictive learning of the

lightweight video-based student model from a frozen *image-based teacher*. We adopt a frozen Mo-bileCLIP (Vasu et al., 2024) as the teacher and align the student's features with temporally shifted teacher's features, enabling it to anticipate future visual cues without requiring a temporally aware teacher. This model-agnostic, data-efficient strategy is particularly well-suited for low-data scenarios.

**Teacher-Student Alignment.** During distillation, our framework leverages the multi-modal MM-AU dataset (Fang et al., 2024), which provides dashcam accident videos paired with accident-related textual descriptions, and supplements it with non-accident videos from Nexar (Moura et al., 2025). In the spatial encoding stage, the student aligns its feature representations with those of the frozen teacher at the same time step. For each stage $\ell \in 1, 2, 3$, with $\mathcal{P}_\ell$ denoting a projection layer, the spatial distillation loss is defined as:

$$\mathcal{L}_{\text{spatial}} = \sum_{\ell=1}^{3} \left|\left| \mathcal{P}_\ell(f_{t,\ell}^{(S)}) - f_{t,\ell}^{(T)} \right|\right|_2^2, \tag{11}$$

where $f_{t,\ell}^{(S)}$ and $f_{t,\ell}^{(T)}$ denote the student's and teacher's features at stage $\ell$ and time $t$, respectively.

In the temporal layers, the student is supervised to predict the teacher's future frame features at time $t+1$ based on its own features at time $t$. This setting introduces a temporally shifted supervision signal, encouraging the student to learn predictive temporal representations. The temporal distillation loss is formulated as:

$$\mathcal{L}_{\text{temporal}} = \left|\left| \mathcal{H}_{\text{ST}}(f_t^{(S)}) - f_{t+1}^{(T)} \right|\right|_2^2, \tag{12}$$

where $\mathcal{H}_{\text{ST}}$ denotes a spatio-temporal projection head applied to the student's output $f_t^{(S)}$ at time $t$, and $f_{t+1}^{(T)}$ is the teacher's spatial feature map at time $t + 1$.

**Contrastive Supervision.** We further apply a CLIP-style contrastive loss between student video features and accident-related text prompts (e.g., "a car runs a red light"), grounding features in semantic accident categories:

$$\mathcal{L}_{\text{contr.}} = -\frac{1}{2B} \sum_{i=1}^{B} \left[ \log \frac{\exp\left(\text{sim}(x_i^{(S)}, z_i^{(S)})/\tau\right)}{\sum_{j=1}^{B} \exp\left(\text{sim}(x_i^{(S)}, z_j^{(S)})/\tau\right)} + \log \frac{\exp\left(\text{sim}(z_i^{(S)}, x_i^{(S)})/\tau\right)}{\sum_{j=1}^{B} \exp\left(\text{sim}(z_i^{(S)}, x_j^{(S)})/\tau\right)} \right], \tag{13}$$

where $B$ is the batch size, $x_i^{(S)}$ and $z_i^{(S)}$ denote the student's visual and textual embeddings, $\text{sim}(\cdot, \cdot)$ is the cosine similarity, and $\tau$ is a learnable temperature parameter. This formulation encourages alignment of matched pairs while separating mismatched ones, enhancing generalization to diverse accident scenarios.

**Overall Objective Function.** A linear classifier with an anticipation loss (Jain et al., 2016) facilitates early accident detection via an exponentially weighted cross-entropy:

$$\mathcal{L}_{\text{accident}} = -\sum_{t=1}^{T} \left[ e^{-\max(0, \frac{T_y-t}{f})} \log\left(a_t^{(p)}\right) + \log(1 - a_t^{(n)}) \right], \tag{14}$$

where $T_y$ is the accident start frame, $f$ the frame rate, $a_t^{(p)}$ and $a_t^{(n)}$ represent accident and non-accident scores. The final training objective is a weighted sum of all components:

$$\mathcal{L}_{\text{total}} = \lambda_1 \mathcal{L}_{\text{spatial}}^{\text{distill}} + \lambda_2 \mathcal{L}_{\text{temporal}}^{\text{distill}} + \lambda_3 \mathcal{L}_{\text{contr.}} + \lambda_4 \mathcal{L}_{\text{accident}}, \tag{15}$$

with $\lambda_i$ controlling their relative importance.

# 4 EXPERIMENTS

## 4.1 EXPERIMENTAL SETUPS

**Benchmark datasets.** We evaluate on widely used benchmarks. **DAD** (Chan et al., 2016) contains 1,750 5-second dashcam videos (20 FPS), with 620 accident and 1,130 non-accident cases. Accidents always occur in the last 0.5 seconds. **CCD** (Bao et al., 2020) includes 4,500 5-second videos

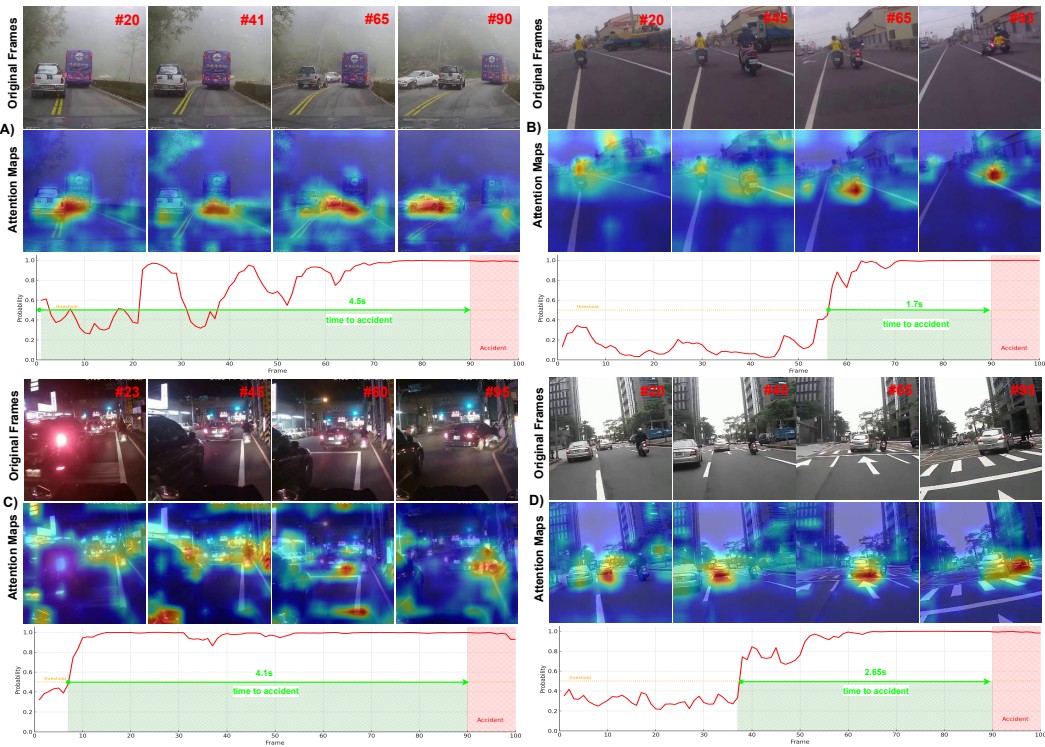

Figure 4: **Qualitative results of accident anticipation under diverse driving conditions.** Red regions in the attention maps indicate areas identified as high risk (e.g., overtaking vehicles, pedestrians, or brake lights). The bottom row shows frame-wise accident probability curves, where earlier rises correspond to successful early anticipation. These examples illustrate the model's ability to focus on critical cues across challenging scenarios such as fog, nighttime driving, and sudden lane changes.

(10 FPS) under diverse driving conditions, with an accident-to-non-accident ratio of 1:2. Full dataset statistics and splits are provided in the supplementary.

**Evaluation Metrics.** Following common evaluation protocol (Chan et al., 2016), we report mean Average Precision (mAP) and mean Time-to-Accident (mTTA). mAP is computed from frame-level accident confidence scores over varying thresholds. mTTA measures the average time between a correct prediction and the accident onset. Formal definitions are included in the supplementary.

**Implementation Details.** Our approach follows a two-stage pipeline: pre-training with vision-language distillation on MM-AU and Nexar, followed by fine-tuning on DAD and CCD. Training uses the AdamW optimizer with a learning rate of $8 \times 10^{-5}$, cosine decay, weight decay of 0.01, and gradient clipping at 5.0. The student processes 8-frame clips during pre-training and 32-frame clips during fine-tuning at an input resolution of $224 \times 224$. Pre-training leverages 112 accident-related prompts, whereas fine-tuning uses binary accident labels. During inference, the model processes 32-frame windows with half overlap and aggregates via max pooling. Additional hyper-parameters and architectural details are provided in the Appendix.

## 4.2 EDGE DEVICE DEPLOYMENT

We train our model on a server equipped with an NVIDIA RTX 4090 (24 GB) and deploy it on an embedded Jetson Orin Nano (8 GB). To optimize for edge deployment, we prune redundant PyTorch operations (e.g., reshaping) prior to exporting the model to ONNX. The model is then compiled with TensorRT (Ubuntu 22.04, JetPack 6.2, TensorRT 10.3) using BF16 precision, yielding a final model size of under 69 MB (excluding the text encoder, as text embeddings are pre-computed and cached during inference).

Table 1: Impact of spatio-temporal RWKV.

| RWKV Layers | Params. (M) | mAP (%) | mTTA (s) |
|---|---|---|---|
| 2 | 13.0 | 71.9 | 3.81 |
| 4 | 19.9 | 73.9 | 3.90 |
| **6** | 26.7 | **75.33** | **4.04** |
| 8 | 33.6 | 71.6 | 4.05 |

Table 2: Impact of distillation components.

| $\mathcal{L}_{spatial}^{Distill}$ | $\mathcal{L}_{temporal}^{Distill}$ | $\mathcal{L}_{contr.}$ | mAP (%) | mTTA (s) |
|---|---|---|---|---|
| ✓ | ✓ | - | 70.1 | 3.54 |
| ✓ | - | ✓ | 71.2 | 3.79 |
| - | ✓ | ✓ | 74.1 | 3.95 |
| ✓ | ✓ | ✓ | **75.3** | **4.04** |

We evaluate performance in terms of both latency and FPS. Latency is defined as the time required to process an initial batch of 32 frames, while FPS is calculated by dividing the total number of processed frames by the total runtime. With batch processing of 32 frames with half overlap, our optimized implementation achieves an inference speed of 80 FPS, corresponding to a latency of approximately 0.4 seconds – well suited for real-time deployment on 30 FPS video streams. During inference, only the vision encoder and classifier head remain active, as text embeddings are cached beforehand.

## 4.3 ABLATION STUDY

To analyze our design choices, we ablate three factors: distillation components, block depth, and mask ratio. Results show that each component contributes to learning effectiveness, deeper blocks balance accuracy and anticipation with complexity, and moderate masking improves robustness without harming performance.

**Impact of Spatio-Temporal RWKV Blocks.** Table 1 evaluates the impact of varying the number of spatio-temporal RWKV layers. Performance improves consistently up to 6 layers, which yields the highest mAP (75.33%) and a strong mTTA (4.04s) with only 26.7M parameters. Increasing to 8 layers adds complexity but slightly reduces performance, likely due to overfitting, suggesting that 6 layers provide the optimal balance between accuracy, temporal reasoning, and model efficiency.

**Impact of Distillation Components.** Table 2 presents an ablation study evaluating the contribution of each component in our distillation framework. The accident loss from Equation 14 is used by default. Removing the contrastive loss leads to a performance drop (from 75.3% to 70.1% mAP), as the downstream fine-tuning stage still leverages the text-visual output – thus depending on the alignment established during pre-training. Temporal and spatial distillation offer complementary benefits: using only temporal supervision achieves better performance (74.1%) than using spatial alone (71.2%), highlighting the effectiveness of our proposed temporally shifted distillation. When combined with contrastive learning, all components work synergistically to yield the best results – 75.3% mAP and 4.04s mTTA – demonstrating the value of joint multi-objective distillation.

**Impact of Masking Strategy.** Table 3 reports the impact of varying spatio-temporal masking ratios in RWKV blocks during pre-training. Performance improves with moderate masking, peaking at 30% (mAP 75.3%, mTTA 4.04s), indicating that partial input encourages robust context learning. However, higher masking ratios (50–75%) degrade performance, suggesting that excessive information removal hinders accurate anticipation.

Table 3: Impact of the masking ratio for spatio-temporal RWKV blocks.

| Masking Ratio (%) | mAP (%) | mTTA (s) |
|---|---|---|
| No mask | 74.0 | 4.00 |
| 30 | **75.3** | **4.04** |
| 50 | 71.8 | 3.93 |
| 75 | 71.0 | 3.94 |

**Impact of Temporal Shift in Distillation.** Table 4 analyzes how different temporal-shift values affect accident-anticipation performance. A moderate shift of one frame yields the best overall trade-off, achieving the highest mAP (75.3%) together with a strong mTTA of 4.04 s. Increasing the shift to two frames results in a slight reduction in mAP (74.7%) while offering only a marginal gain in mTTA (4.06 s). A larger three-frame shift further decreases mAP (70.7%) despite a small improvement in mTTA (4.13 s). These results indicate that although larger temporal shifts promote earlier prediction, they come at the cost of reduced accuracy. Consequently, a small temporal shift provides the most effective balance and aligns with the short predictive horizon characteristic of accident-anticipation tasks.

Table 4: Impact of the various temporal shift in temporally shifted distillation.

| Temporal Shift | mAP (%) | mTTA (s) |
|---|---|---|
| no shift (only spatial) | 71.2 | 3.79 |
| 1 | **75.3** | 4.04 |
| 2 | 74.7 | 4.06 |
| 3 | 70.7 | **4.13** |

Table 5: Impact of a video-based teacher in temporally shifted distillation.

| Video Distillation | mAP (%) | mTTA (s) |
|---|---|---|
| VJEPA2 – no shift | 65.3 | 4.12 |
| VJEPA2 – shift (1) | **66.0** | **4.21** |

**Impact of Video Teacher.** We evaluate a video-based teacher using V-JEPA2 Assran et al. (2025), a recent state-of-the-art model for spatio-temporal representation learning. As shown in Table 5, introducing a temporal shift continues to yield meaningful benefits: the anticipation time increases from 4.12 s to 4.21 s, demonstrating that our shifted distillation remains effective even when the teacher already encodes temporal dynamics. However, the overall mAP achieved with V-JEPA2 is notably lower than that obtained with the image-based MobileCLIP teacher. This performance gap is expected due to architectural mismatches – V-JEPA2 employs different spatial tokenization and operates at a much lower spatial resolution in its shallow layers, making its spatial representations difficult for the student to align with. Consequently, the learning process is driven primarily by temporal supervision rather than spatial correspondence. While the temporal shift still improves early anticipation, the lack of compatible high-resolution spatial guidance results in reduced mAP compared to the MobileCLIP-based setting.

**Impact of Temporal RWKV and Distillation.** Table 6 highlights the complementary roles of Temporal RWKV (T-RWKV) and Temporally Shifted Distillation (TSD). Using T-RWKV alone with fine-tuning (FN) provides limited benefit, achieving only 39.4% mAP. In contrast, applying TSD with the purely Spatial RWKV (S-RWKV) blocks yields a substantial improvement to 55.6% mAP and mTTA of 4s, indicating that shifted supervision supplies

Table 6: Ablation of Temporal RWKV (T-RWKV) and Temporally Shifted Distillation (TSD).

| Module | mAP (%) | mTTA (s) |
|---|---|---|
| T-RWKV + FN | 39.4 | 3.97 |
| S-RWKV + TSD + FN | 55.6 | 4.00 |
| T-RWKV + pre-train + FN | 67.7 | 4.00 |
| T-RWKV + TSD + FN | **75.3** | **4.04** |

meaningful future-aware cues even without temporal recurrence. However, effective temporal modeling is essential for fully exploiting this signal: T-RWKV with pre-training reaches 67.7% mAP, and the combination of T-RWKV and TSD achieves the best performance at 75.3% mAP with an mTTA of 4.04 s. These results demonstrate that neither TSD nor temporal modeling alone is sufficient –strong anticipatory capability arises only when both forms of supervision are jointly employed.

## 4.4 QUALITATIVE RESULTS

Figure 4 illustrates successful cases of early accident anticipation. In A), a vehicle collides while overtaking into the opposite lane under foggy conditions; the model maintains focus on both the bus and the overtaking car, achieving a 4.5s early warning despite limited visibility. In B), a motorcyclist crashes into a turning vehicle; the model sharply attends to the risky motion around the motorcycle and triggers a 1.7s advance alert. C) presents a nighttime rear-end collision, where attention is concentrated on brake lights and dense traffic, allowing a 4.1s prediction. In D), a sudden lane switch results in a side collision; the model progressively attends to the intruding vehicle and predicts the accident 2.6s ahead of time.

## 4.5 STATE-OF-THE-ART COMPARISON

All previous works rely on object detection (Faster-RCNN) and external feature extraction pipelines (VGG-16), introducing significant computational overhead and latency. While MASTTA is the only other recent end-to-end approach, our method achieves higher mAP (+5.1%) and mTTA (+0.08s), while being $3.8\times$ smaller – enabling more efficient early accident prediction.

Table 7 provides a comprehensive comparison of recent methods on the DAD and CCD datasets. On DAD, our model delivers the best balance between early anticipation (mTTA of 4.04s) and high precision (75.33% mAP), outperforming all prior approaches in the trade-off between these metrics.

Table 7: Comparison of the longest mTTA and the corresponding mAP of SOTA models on the DAD and CCD. **Best balanced results** in terms of model size, accuracy, and early prediction are in bold.

| Dataset | Method | Source | Inference Params. (M) | mAP(%) | mTTA(s) |
|---------|--------|--------|-----------------------|--------|---------|
| DAD | DSA Chan et al. (2016) | ACCV16 | 179 | 48.4 | 1.34 |
| | adaLEA Suzuki et al. (2018) | CVPR18 | 180 | 52.3 | 3.43 |
| | GCRNN Bao et al. (2020) | ACMMM20 | 275 | 53.7 | 3.53 |
| | FA Fatima et al. (2021) | ICPR21 | 78 | 49.8 | 3.76 |
| | DRIVE Bao et al. (2021) | ICCV21 | 140 | 62.8 | 2.78 |
| | L-RA Zeng et al. (2017) | CVPR21 | 185 | 49.1 | 3.04 |
| | DSTA Karim et al. (2022) | TITS22 | 180 | 56.1 | 3.66 |
| | GSC Wang et al. (2024) | TIV23 | 75 | 60.4 | 2.55 |
| | DAA-GNN Song et al. (2024) | PR23 | 183 | 70.6 | 1.59 |
| | MASTTA Patera et al. (2025) | TCSVT25 | 99 | 70.2 | 3.96 |
| | CCAF-Net Liu et al. (2025) | NEURO25 | 191 | 71.8 | 4.15 |
| | **Ours** | - | **26** | **75.3** | **4.04** |
| CCD | DSA Chan et al. (2016) | ACCV16 | 179 | 99.6 | 4.87 |
| | GCRNN Bao et al. (2020) | ACMMM20 | 275 | 99.5 | 4.74 |
| | DSTA Karim et al. (2022) | TITS22 | 180 | 99.6 | 4.52 |
| | MASTTA Patera et al. (2025) | TCSVT25 | 99 | 99.9 | 4.95 |
| | CCAF-Net Liu et al. (2025) | NEURO25 | 191 | 93.9 | 4.94 |
| | **Ours** | - | **26** | **99.9** | **4.95** |

Notably, this is achieved with only 26M parameters – $7\times$ fewer than DAA-GNN (183M), $8.3\times$ fewer than CCAF-Net (191M), and $3.8\times$ fewer than MASTTA (99M). On CCD, our model achieves the highest mAP (99.9%) and the longest mTTA (4.95s), matching MASTTA and exceeding other SOTA, again using a fraction of the parameters.

Table 8 compares prior methods on the DAD dataset from a high mAP perspective. We additionally incorporate several purely Transformer-based approaches Fan et al. (2021); Liu et al. (2022); Li et al. (2023) as well as CNN-based models Carreira & Zisserman (2017); Tran et al. (2018); Lin et al. (2019), all of which are widely used in video understanding. Notably, these architectures underperform on the accident anticipation task, with none exceeding 70% mAP. Although CCAF-Net and MASTTA slightly outperform in mAP (81.3% and 80.8%, respectively), our model attains a competitive 79.61% mAP with the smallest parameter count (26M) and a favorable mTTA of 3.41s – outperforming MASTTA by 0.09s and trailing CCAF-Net by only 0.34s. These results underscore our model's strong efficiency–accuracy trade-off, making it well-suited for real-time accident anticipation.

Table 8: Comparison of the highest mAP and corresponding mTTA of prior models on DAD. **Best balanced results** in terms of model size, accuracy, and early prediction are in bold.

| Method | Inference Params. (M) | mAP(%) | mTTA(s) |
|--------|-----------------------|--------|---------|
| I3D | 21 | 68.0 | 2.99 |
| R(2+1)D | 31 | 49.7 | 3.57 |
| TSM | 43 | 53.0 | 3.39 |
| GCRNN | 275 | 72.2 | 1.33 |
| L-RAI | 185 | 51.4 | 3.01 |
| MVITv2 | 51 | 64.4 | - |
| DSTA | 180 | 72.3 | 1.50 |
| VideoSwin | 88 | 65.4 | - |
| UniFormer V2 | 115 | 65.2 | - |
| DAA-GNN | 183 | 75.2 | 1.47 |
| MASTTA | 99 | 80.8 | 3.32 |
| CCAF-Net | 191 | 81.3 | 3.75 |
| **Ours** | **26** | **79.6** | **3.41** |

## 5 CONCLUSIONS

This work presents a lightweight spatio-temporal framework for early accident anticipation, designed for real-time operation in resource-constrained environments. By combining a temporally shifted distillation strategy with a hybrid architecture that integrates convolutional spatial encoding and recurrent temporal modeling, our approach enables efficient long-range reasoning without the need for video-trained teachers. The framework achieves strong accuracy on limited data with low latency and memory usage, making it practical for deployment in real-world autonomous driving systems and adaptable to broader traffic safety applications.

ACKNOWLEDGMENTS

This work was supported by the National Science and Technology Council, R.O.C., under contracts 113-2221-E-011-132-MY2 and 114-2221-E-011-072.

REPRODUCIBILITY STATEMENT

All architectural details and parameter settings are described in the Implementation Details section, with further configuration and training specifics provided in the supplementary material in Appendix to facilitate reproducibility. The project page and code available at `https://patpatera.github.io/TSD-Accident-Anticipation`.

The manuscript's writing has been polished using ChatGPT.

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

## A    SUMMARY OF SUPPLEMENTARY CONTRIBUTIONS

This supplementary material provides additional details supporting our main submission. We describe the proposed spatio-temporal masking strategy along with ablation results, present the complete architectural configurations of both the student and teacher models, and outline the training protocols and deployment setup. We also include qualitative examples that highlight model limitations, clarify the dataset composition, and explain the use of coarse-grained text labels during fine-tuning. The project page and code available at `https://patpatera.github.io/TSD-Accident-Anticipation`.

## B    ANALYSIS OF CCD PERFORMANCE

To further address concerns regarding potential overfitting on the CCD dataset, we provide additional analyses and metrics beyond the mAP and mTTA reported in the main paper. CCD is a well-known saturated benchmark where numerous prior works (e.g., DSA, GCRNN, MASTTA) routinely achieve 99.0–99.9% mAP under the standard protocol, and our results follow this established trend. To verify robustness, Table 9 reports ROC-AUC, which reaches a similarly high value of 85.0%, indicating strong separability between accident and non-accident trajectories. We also evaluate a smaller student variant (XS) with only 13 million parameters; despite its lower capacity, it preserves the same mAP of 99.9% with only a minor decrease in mTTA (4.92 s), suggesting that CCD performance does not depend on fragile model capacity. Furthermore, we include $TTA_{80\%}$ – the time-to-accident at 80% recall – which remains consistently high at 4.84 s. Collectively, these results confirm that our model's CCD performance is stable and reflects the dataset's inherent ease rather than overfitting.

Table 9: Comparison of smaller network variant on CCD with additional metrics.

| Model | AUC (%) | mAP (%) | mTTA (s) | TTA$_{80\%}$ | Params (M) |
|---|---|---|---|---|---|
| Model - smaller (XS) | 85.0 | 99.9 | 4.92 | 4.84 | 13 |
| Model - small (S) | 86.9 | 99.9 | 4.95 | 4.86 | 26 |

## C   ANALYSIS OF SHIFTED-TEACHER ALIGNMENT

We further assess the effectiveness of Temporally Shifted Distillation (TSD) by examining the difference in cosine similarity between two pairs of signals: the student's features $X_t^S$ aligned with the teacher's same-time features $X_t^T$, and the student's features $X_t^S$ aligned with the teacher's future features $X_{t+1}^T$. To further quantify this effect, we introduce the *Temporal Alignment Ratio* (TAR),

$$\text{TAR}_t = \frac{\cos\big(X_t^S, X_{t+1}^T\big)}{\cos\big(X_t^S, X_t^T\big) + \varepsilon}, \tag{16}$$

which measures how strongly the student aligns with the teacher's *future* state relative to its *present* state. Values greater than 1 indicate that the student preferentially aligns with future representations, reflecting true anticipatory capability rather than static feature matching. Empirically, TAR remains consistently above 1 and increases over time, further confirming that TSD induces meaningful temporal forecasting behavior. As shown in Fig. 5, a decreasing blue TAR curve (smooth) would indicate that TSD provides no benefit, suggesting that the student relies primarily on spatial correlations rather than learning to anticipate future representations. In contrast, the consistently increasing trend demonstrates that the student becomes progressively better aligned with the *future* teacher embeddings.

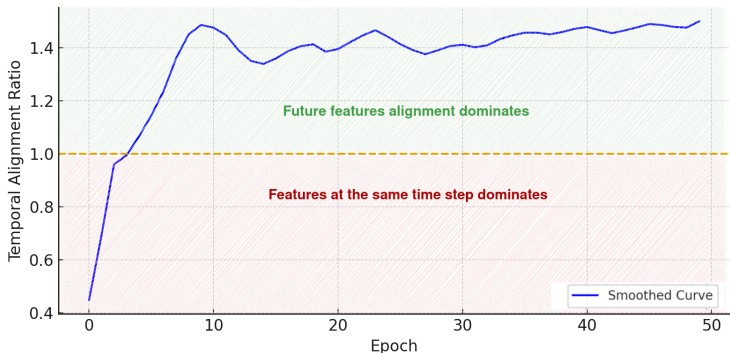

Figure 5: Temporal alignment comparison showing that the student aligns more strongly with the teacher's *future* features. TAR (blue curve) $> 1$ further confirms TSD induces anticipatory learning.

## D   ARCHITECTURAL DETAILS

### D.1   STUDENT VISION ENCODER

Our Student Vision Encoder adopts a lightweight yet expressive RepMixer (illustrated in Figure 6) spatial encoding on the MobileCLIP Vasu et al. (2024) with our lightweight window-based spatio-temporal module adapted from the RWKV Peng et al. (2023), optimized for efficient spatio-temporal modeling in accident anticipation. As detailed in Table 10, the encoder processes an input video of shape $T \times H \times W$ through four hierarchical stages, each progressively reducing spatial resolution while expanding feature dimensionality. The stem consists of two convolutional layers – a standard $3 \times 3$ convolution followed by a MobileOne-style $3 \times 3$ convolution with stride 2 – yielding 48-dim features.

Stage 1 begins with a patch embedding layer using a $7 \times 7$ MobileOne-style convolution with stride 2 to generate tokens at $\frac{H}{4} \times \frac{W}{4}$ resolution, followed by 2 RepMixer blocks with 96-dim features

Table 10: **Student Vision Encoder.** Architectural configuration of the proposed lightweight spatio-temporal model variant.

| Stage | #Tokens | Layer Spec. | Description | Dim. |
|---|---|---|---|---|
| Stem | $T \times H \times W$ | Conv | $3 \times 3$, stride 2 
 $3 \times 3$ MobileOne Style, stride 2 | 48 |
| 1 | $T \times \frac{H}{4} \times \frac{W}{4}$ | Patch Embed. | $7 \times 7$ MobileOne Style, stride 2 | 96 |
| | | Blocks | RepMixer 
 2× | |
| 2 | $T \times \frac{H}{8} \times \frac{W}{8}$ | Patch Embed. | $7 \times 7$ MobileOne Style, stride 2 | 128 |
| | | Blocks | RepMixer 
 6× | |
| 3 | $T \times \frac{H}{16} \times \frac{W}{16}$ | Patch Embed. | $7 \times 7$ MobileOne Style, stride 2 | 256 |
| | | Blocks | RepMixer 
 10× | |
| 4 | $T \times \frac{H}{32} \times \frac{W}{32}$ | Patch Embed. | $7 \times 7$ MobileOne Style, stride 2 | 512 |
| | | Blocks | **Spatio-temporal RWKV** 
 6× | |
| | | Parameters (M) | | 26 |

for local spatial modeling. Stage 2 continues this pattern, reducing the resolution to $\frac{H}{8} \times \frac{W}{8}$, and applies 6 RepMixer blocks with 128 channels. Stage 3 further downsamples to $\frac{H}{16} \times \frac{W}{16}$ and employs 10 RepMixer blocks with 256-dim features to capture mid-level semantic representations.

In Stage 4, the spatial resolution is reduced to $\frac{H}{32} \times \frac{W}{32}$, resulting in $7 \times 7$ spatial tokens with input of $224 \times 224$. Here, 3D conditional positional encoding is applied, and temporal modeling is introduced using 6 Spatio-temporal RWKV blocks with 512-dim tokens. These RWKV blocks allow efficient linear-complexity modeling across frames, enabling the encoder to capture long-range temporal dependencies without sacrificing latency. The complete Vision Encoder comprises only 26 million parameters.

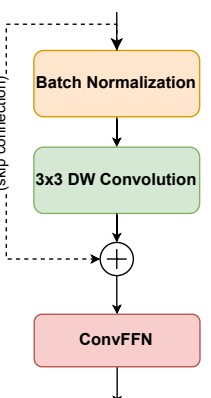

Figure 6: **RepMixer Block from MobileCLIP.**

### D.2 STUDENT TEXT ENCODER

The text encoder adopts the Text-RepMixer architecture from MobileCLIP, maintaining architectural consistency with the vision encoder while being specifically tailored for language processing. It employs a hybrid design that combines 1D convolutions with self-attention layers. Unlike purely convolutional approaches – which were found to underperform – Text-RepMixer effectively balances local and global context modeling by integrating convolutional token mixing with attention-based reasoning. For efficient inference, skip connections and normalization layers are reparameterized, and the feed-forward layers are enhanced with depthwise 1D convolutions, forming ConvFFN blocks that can be fused for optimized execution. The complete model, including both the Vision and Text Encoders, contains a total of 69 million parameters.

### D.3 TEACHER VISION ENCODER

The Teacher Vision Encoder shares the same overall architecture as the Student, detailed in Figure 11, including the hierarchical four-stage structure and patch embedding. However, it is deeper, with [4, 12, 24, 4] blocks and larger embedding dimensions of [80, 160, 320, 640] across stages 1 to

Table 11: **Teacher Vision Encoder**. Architectural configuration of the large image-based model variant serving as the teacher.

| Stage | #Tokens | Layer Spec. | Description | Dim. |
|---|---|---|---|---|
| Stem | $T \times H \times W$ | Conv | $3 \times 3$, stride 2 
 $3 \times 3$ MobileOne Style, stride 2 | 48 |
| 1 | $T \times \frac{H}{4} \times \frac{W}{4}$ | Patch Embed. | $7 \times 7$ MobileOne Style, stride 2 | 80 |
| | | Blocks | RepMixer 
 $4\times$ | |
| 2 | $T \times \frac{H}{8} \times \frac{W}{8}$ | Patch Embed. | $7 \times 7$ MobileOne Style, stride 2 | 160 |
| | | Blocks | RepMixer 
 $12\times$ | |
| 3 | $T \times \frac{H}{16} \times \frac{W}{16}$ | Patch Embed. | $7 \times 7$ MobileOne Style, stride 2 | 320 |
| | | Blocks | RepMixer 
 $24\times$ | |
| 4 | $T \times \frac{H}{32} \times \frac{W}{32}$ | Patch Embed. | $7 \times 7$ MobileOne Style, stride 2 | 640 |
| | | Blocks | Multi-Head Self-Attention 
 $4\times$ | |
| Parameters (M) | | | | 36 |

4. While the Student introduces lightweight spatio-temporal RWKV blocks in Stage 4, the Teacher retains conditional positional encoding and standard spatial multi-head self-attention (MHSA) for purely spatial modeling. The Teacher is initialized with the official pre-trained weights from Mobile-CLIP Vasu et al. (2024) and kept entirely frozen during pre-training, providing stable, high-capacity supervision for the student's spatio-temporal adaptation via our proposed temporally shifted distillation.

| Hyper-parameter | Pre-training | Fine-tuning |
|---|---|---|
| Input resolution | 224×224 | 224×224 |
| Input #frames | 8 | 32 |
| Frame sample interval | 8 | 3 |
| Random Resize Crop | [0.1, 1.0] | [0.1, 1.0] |
| Random Horizontal Flip | 0.5 | 0.5 |
| Color Jitter | ✓ | ✓ |
| $\lambda_1, \lambda_2, \lambda_3, \lambda_4$ | [0.3, 0.4, 0.5, 1.0] | - |
| Mask ratio | 30% | - |
| Train epochs | 70 | 50 |
| Warmup epochs | 5 | 0 |
| Batch size | 70 | 12 |
| Optimizer | AdamW | AdamW |
| Peak learning rate | 8e-5 | 8e-6 |
| LR decay schedule | cosine | cosine |
| Weight decay rate | 0.01 | 0.01 |
| Gradient clipping | 5 | 5 |
| Mixed precision | BF16 | BF16 |

Table 12: Hyperparameters for Pre-training and Training.

# E    EXPERIMENTAL SETUPS

We conducted all experiments using a single NVIDIA RTX 4090 GPU with 24 GB memory for both pre-training and fine-tuning, while the final model was deployed on an NVIDIA Jetson Orin Nano with 8 GB shared memory for real-time inference. The pre-training and fine-tuning phase have utilized the dataset described in the Datasets section of this supplementary materials. Table 12

summarizes the hyper-parameter configurations used for both pre-training and fine-tuning. In both settings, the input resolution is fixed at $224 \times 224$. During pre-training, the model processes clips of 8 frames sampled every 8 frames, while fine-tuning uses 32-frame clips sampled every 3 frames to improve temporal precision for accident anticipation. For input data augmentation, both phases employ random resize cropping (scale range 0.1 - 1.0), horizontal flipping with a probability of 0.5, and color jitter is applied to promote visual robustness.

The pre-training phase additionally incorporates a 30% masking ratio applied to the input frames and employs a multi-task loss with weighting coefficients $\lambda_1, \lambda_2, \lambda_3, \lambda_4 = [0.3, 0.4, 0.5, 1.0]$, whereas fine-tuning relies solely on the classification loss. Both phases are trained for 50 epochs, with 5 warm-up epochs. Due to memory constraints, the batch size is set to 70 during pre-training and 12 during fine-tuning. Optimization is performed using AdamW with a peak learning rate of $8 \times 10^{-5}$ and cosine learning rate decay schedule. A weight decay rate of 0.01 and gradient clipping threshold of 5 are consistently applied throughout both phases. Both pre-training and training were performed under PyTorch mixed precision `BF16`. Training across multiple random seeds, mAP remains stable within $\pm 0.3\%$ and mTTA within $\pm 0.08$ s.

Fine-tuning uses the same objective function $\mathcal{L}_{\text{accident}}$ as defined in Equation (14) of the main manuscript for text-visual features, along with a standard cross-entropy loss for the visual classifier.

## F  EVALUATION METRICS

**Average Precision (AP).** Each method produces a confidence score per frame representing the likelihood of a future accident. A prediction is counted as a True Positive (TP) if the score exceeds a threshold $q$ before the accident onset; otherwise, it is treated as a False Negative (FN). Any prediction in a non-accident video is considered a False Positive (FP). By varying $q$, a precision-recall curve is obtained, and AP is computed as the area under this curve. Precision ($\text{TP}/(\text{TP} + \text{FP})$) and recall ($\text{TP}/(\text{TP} + \text{FN})$) are evaluated at each threshold, and the mean Average Precision (mAP) is calculated as their average across all evaluated thresholds.

**Time to Accident (TTA).** TTA is defined as $T - \tau$, where $T$ is the accident start frame and $\tau$ is the first frame where the confidence exceeds $q$. The mean TTA (mTTA) averages TTA over true positives across thresholds.

## G  PERFORMANCE

Figure 7 illustrates the trade-off between model size and early accident anticipation ability across state-of-the-art methods. The *x*-axis represents the number of inference parameters (in millions), and the *y*-axis denotes the mean Time-to-Accident (mTTA) in seconds. Each point corresponds to a competing approach. Our method, highlighted in green and marked with a red arrow, achieves the highest mTTA (4.04s) while using only 26M parameters—the smallest among all models. This result demon-

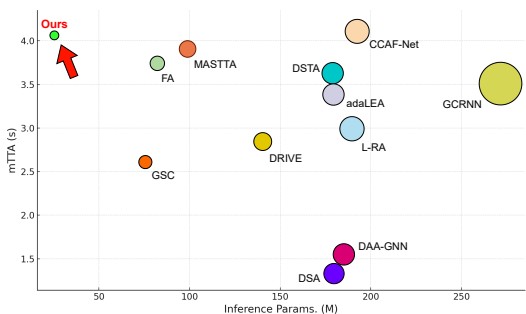

Figure 7: **Inference Parameters vs. Early Prediction (mTTA).** Our method (green) achieves real-time performance with the smallest model size, surpassing larger baselines in terms of mTTA. This demonstrates a favorable trade-off between efficiency and temporal anticipation, ideal for resource-limited deployment.

strates that our approach delivers superior early anticipation despite being significantly more lightweight than existing baselines, offering a clear advantage for efficient, real-world deployment.

## H  IMPLEMENTATION DETAILS

We implement our model and all training/testing pipelines using PyTorch 2.5.0. To enable efficient spatio-temporal modeling, our RWKV-based module is implemented with a custom CUDA kernel that supports parallel computation over video spatio-temporal input sequence, compiled with Ninja

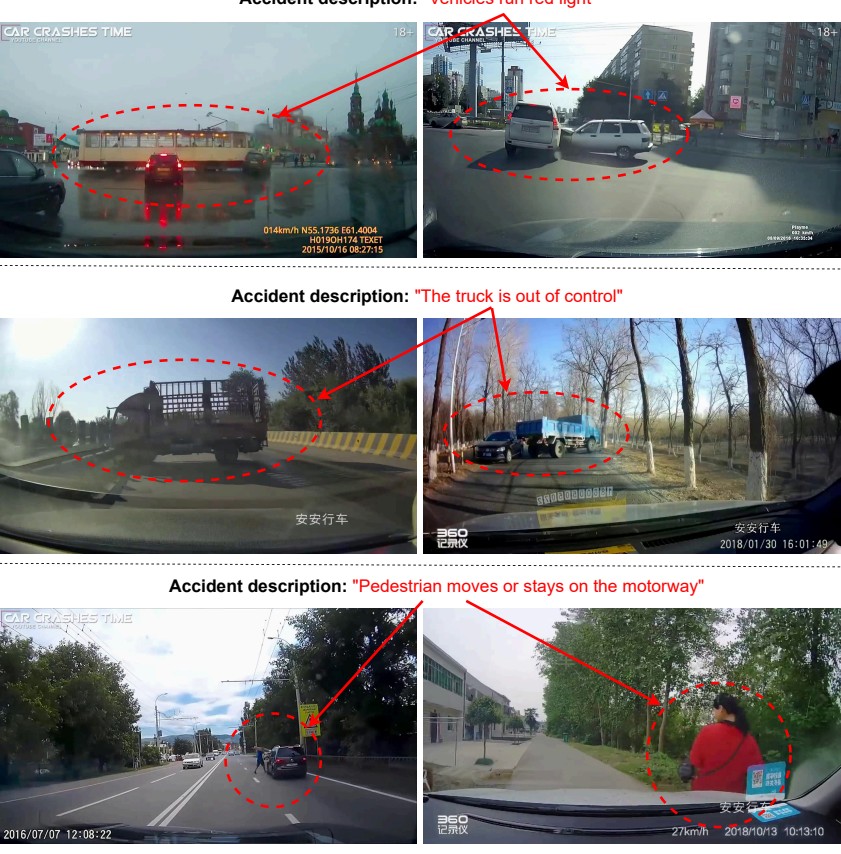

Figure 8: Example of accident scenes from MM-AU with corresponding textual descriptions used during pre-training.

1.10.2.1 library . This design significantly accelerates both training and inference, particularly when handling long video sequences. To prepare window-based inputs for the RWKV block, we utilize PyTorch's built-in nn.Unfold operation to window-based extract spatio-temporal patches, and nn.Fold to reconstruct the processed output back to the original input shape. This approach ensures computational efficiency while maintaining compatibility with the rest of the architecture. The similar unfolding window-based pattern can be found in another methods such as SWIN-Transformer Liu et al. (2022) and MobileViT Mehta & Rastegari (2021).

# I DATASETS

The training of our framework involves both pre-training and fine-tuning phases. We pre-train the model on the recently released MM-AU Fang et al. (2024) dataset, which includes only accident videos paired with textual descriptions detailing 112 distinct accident scenarios. To compensate for the absence of negative samples, we incorporate non-accident videos from Nexar Moura et al. (2025), DAD Chan et al. (2016), and CCD Bao et al. (2020). This setup offers key benefits: (1) MM-AU's textual descriptions provide strong inductive bias, helping the model learn causal relationships beyond visual saliency; (2) pre-training on a separate dataset supports domain generalization and reduces overfitting; and (3) exposure to varied accident types, viewpoints, and contexts enhances robustness. MM-AU is not used for evaluation due to its lack of negative samples and standardized test splits with evaluation protocol. Instead, we leverage it solely for pre-training to exploit its rich text–video annotations. Figure 8 shows example accident scenes from MM-AU with their corresponding textual descriptions used during pre-training.

During fine-tuning and inference, we use only coarse-grained text labels — "Normal traffic situation" and "A traffic accident" — due to the absence of detailed textual annotations in the downstream datasets. These labels provide semantic conditioning, enabling class-aware alignment even without fine-grained textual supervision. Fine-tuning is performed on two widely used benchmark datasets for traffic accident anticipation – DAD and CCD – to ensure a fair comparison with prior state-of-the-art methods.

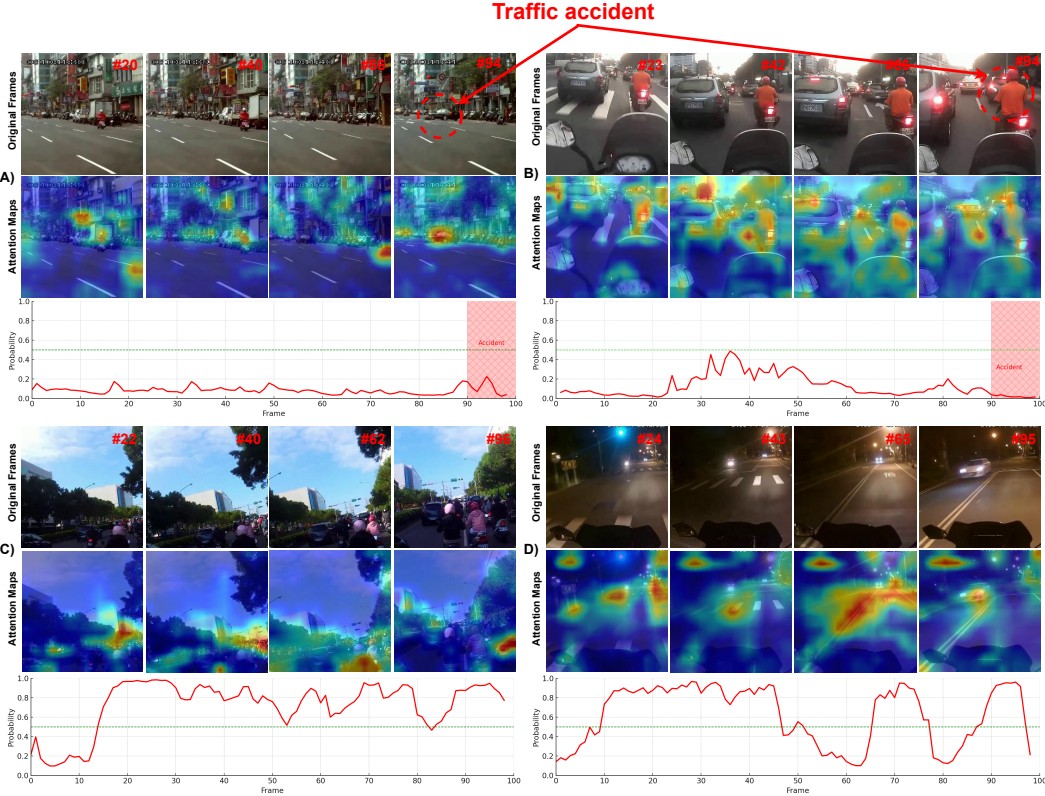

Figure 9: **Qualitative analysis of model limitations.** Examples of false negatives (A–B) and false positives (C–D) are shown. Red regions in the attention maps indicate high-risk areas, while the bottom row displays frame-wise accident probability curves.

## J    LIMITATIONS

Figure 9 presents qualitative results across four diverse driving scenarios, illustrating both false negatives (A–B) and false positives (C–D) in traffic accident anticipation. Subfigures A) and B) depict cases where the model fails to anticipate accidents. In A), the accident occurs in the far background with only a small visual footprint, making it difficult for the model to detect early warning cues. Despite partial attention over relevant regions, the subtlety and scale of the event lead to a missed prediction. In B), the camera is mounted on a motorbike helmet, causing severe camera shake and unstable motion that hinder consistent spatio-temporal reasoning. Although the attention maps correctly highlight the road and nearby vehicles, the predicted accident probability remains low throughout, likely due to the accident being heavily obscured by another vehicle in front.

In contrast, C) and D) represent false positives, where the model erroneously predicts an accident in videos where no incident occurs. Both sequences are captured with helmet-mounted cameras, introducing natural shakiness. C) features dense daytime traffic, with a tightly packed group of motorbikes driving in close proximity. The attention maps strongly focus on surrounding vehicles, causing the model to interpret the scene as hazardous, despite the absence of an accident. D), recorded at night under low visibility, involves high-speed driving with bright headlights and road

reflections. These challenging lighting conditions, combined with motion from the helmet-mounted camera, result in persistently high accident probabilities, even though no collision occurs.

Overall, these visualizations underscore key challenges in accident anticipation, including motion instability, small-scale accident cues, traffic density, and adverse lighting. They highlight the importance of incorporating spatio-temporal consistency, scale awareness, and robustness to visual noise into model design.

