# OpenReview forum: "Lightweight Spatio-Temporal Modeling via Temporally Shifted Distillation for Real-Time Accident Anticipation"
_ICLR.cc/2026/Conference — ICLR 2026 Poster_

### Official Review · Reviewer_bqJU · 2025-10-30

**Soundness:** 3
**Presentation:** 3
**Contribution:** 3
**Rating:** 4
**Confidence:** 2

**Summary:**

The paper introduces a lightweight spatio-temporal framework for real-time traffic accident anticipation, particularly suited for deployment on resource-constrained edge devices. The key contributions are: (1) a novel temporally shifted distillation strategy that enables a student model to learn predictive temporal dynamics from a frozen image-based teacher, eliminating the need for a video pre-trained teacher and making it ideal for small datasets and low-resource settings; (2) a hybrid architecture that combines RepMixer spatial encoding with an RWKV-inspired recurrent module for efficient long-range temporal reasoning, achieving low computational complexity; (3) a masked memory strategy that simulates occlusions and partial visibility to improve robustness under real-world conditions.

**Strengths:**

1. The paper innovatively redefines temporal learning through temporally shifted distillation, removing the need for video-pretrained teachers and enabling real-time performance under constraints.
2. Comprehensive experiments and ablation studies show improvements in mAP and mTTA with fewer parameters.
3. Clear structure, well-explained components, and visual aids make the paper accessible, though some concepts remain high-level.

**Weaknesses:**

1. The stitching method of the spatio-temporal scanning pattern seems somewhat arbitrary. There might be continuous information between feature sub-blocks that is not utilized. Is this an existing design approach?

2. In lines 231-236, I don't understand why only KV is involved in the calculation and not the Q parameter.

3. In lines 241-260, can the masked part be considered multi-stage, requiring multiple training iterations? Does this introduce higher training costs compared to other models? Is this reflected in the paper?

4. Lines 264-269: Are there applications on larger-scale datasets? I would like to know how this lightweight model performs under larger-scale conditions. Even if the performance isn't strong, can you provide the model's applicability boundary?

5. Table 4: The performance on the CCD dataset seems close to overfitting. Has the lightweight version been applied to other methods? Also, are there additional evaluation metrics? Are the two metrics used sufficient?

**Questions:**

See the weakness.

---

> ### Author Response · Authors · 2025-11-21
>
> We appreciate the reviewer’s thoughtful comments. Our detailed responses to the raised concerns are provided below
>
> > 1. The stitching method of the spatio-temporal scanning pattern seems somewhat arbitrary. There might be continuous information between feature sub-blocks that is not utilized. Is this an existing design approach?
>
> Our spatial scanning is not arbitrary; it follows a well-established window-partitioning design similar to Swin Transformer and MobileViT. As detailed in our Supplementary - *"Implementation Details,"* we use nn.Unfold to partition features into fixed-size windows and nn.Fold to reconstruct them back into the global feature map. This preserves spatial continuity and ensures that no information between neighboring windows is lost. In practice, this behaves similarly to standard non-overlapping window attention or convolutional tiling used in prior works.
>
> To further strengthen consistency, our fused CUDA kernel processes all windows in parallel, enabling efficient and local-aware temporal modeling suitable even for edge devices. Importantly, applying RWKV recurrence within localized windows also mitigates the oversmoothing, since temporal mixing occurs only within spatially coherent regions rather than across the entire frame, preserving fine-grained spatial distinctions.
>
> > 2. In lines 231-236, I don't understand why only KV is involved in the calculation and not the Q parameter.
>
> We thank the reviewer for pointing this out. *Channel Mixing* in RWKV is not an attention operation and therefore does not use a $Q$ input. The appearance of only $(K, V)$ in the earlier manuscript was due to a notational carry-over from *Temporal Mixing* block and was misleading.
>
> We have corrected the notation in the revision: *Channel Mixing* uses $(R_t', K_t')$, where $R_t'$ is the gated branch and $K'_t$ is the channel-projection branch.  $(R'_t, K'_t)$ are computed through their respective linear projections $W'_r$ and $W'_k$. The final output is obtained by applying a sigmoid gate $\sigma(R'_t)$ to the squared-ReLU activation of transformed $K'_t$ by $W_o$. Importantly, Channel Mixing does not include a $V$ term; the operation is solely defined by the gated $R'_t$ and $K'_t$. All equations and notation have been revised accordingly for clarity and correctness.
>
>   $$R^\prime_t = W^\prime_r \left( \mu^\prime_r X_t + \left(1 - \mu^\prime_r \right) X_{t-1} \right)$$
>   $$K^\prime_t = W^\prime_k \left( \mu^\prime_k X_t + \left(1 - \mu^\prime_k \right) X_{t-1} \right)$$
>   $$cmix_t = \sigma(R^\prime_t) \odot W^\prime_o \left( ReLU(K^\prime_t)^2 \right)$$
>
> > 3. In lines 241-260, can the masked part be considered multi-stage, requiring multiple training iterations? Does this introduce higher training costs compared to other models? Is this reflected in the paper?
>
> The masked memory mechanism is **not** a multi-stage procedure, nor does it require repeated iterations. It is applied **once per forward pass** during pre-training, exactly like any standard feed-forward operation, and does not involve any additional unrolling or multi-step optimization.
>
> The masking logic is fully implemented within our custom fused CUDA kernel, making the additional cost negligible and keeping the overall training time comparable to baseline RWKV or similar recurrent architectures. During fine-tuning and inference, we use the unmasked variant, which carries no overhead. We have clarified this behavior and its computational implications in Sec. 3.3 of the revised manuscript.
>
> > 4. Lines 264-269: Are there applications on larger-scale datasets? I would like to know how this lightweight model performs under larger-scale conditions. Even if the performance isn't strong, can you provide the model's applicability boundary?
>
> Point taken. As requested, we added results on the large-scale Kinetics-400 dataset to examine the model’s applicability beyond accident anticipation. Due to limited compute resources and time constraints, we were only able to run 40 epochs of TSD pre-training using 8 frames per video. Even under this constrained setting, our model reaches **74.1% top-1 and 92.2% top-5 accuracy**, and continues to improve steadily with more epochs, indicating that the architecture scales effectively to larger datasets and benefits from additional training.
>
> Although this experiment is not meant to compete with fully trained video-recognition models, it shows that our lightweight T-RWKV backbone and shifted-distillation strategy generalize beyond accident anticipation and do not depend on the teacher having temporal capability. Our method is deliberately designed for small- to medium-scale accident datasets, where directly fine-tuning video pre-trained transformers often leads to overfitting and requires computational resources that are impractical for this domain. Instead, our approach provides an efficient and scalable alternative that learns predictive temporal dynamics without relying on expensive video pre-training.

---

> ### Author Response · Authors · 2025-11-21
>
> > 5. Table 4: The performance on the CCD dataset seems close to overfitting. Has the lightweight version been applied to other methods? Also, are there additional evaluation metrics? Are the two metrics used sufficient?
>
> CCD is a well-known saturated benchmark, with many prior state-of-the-art works (e.g., DSA, GCRNN, MASTTA) reporting 99.0 – 99.9% mAP under the official evaluation split and protocol. Our high mAP is consistent with this saturation rather than model overfitting.
>
> To address the concern explicitly, we added additional evaluation metrics and a lightweight student variant:
> - ROC-AUC on CCD, which is also near-saturated at 85.0%, aligns with the trends in earlier works.
> - TTA@80 (time-to-accident at 80% recall rate), which remains high (4.84 s).
> - A smaller student model (XS) with fewer layers still reaches 99.9% mAP with only a slight mTTA reduction (4.92 s).
>
> The consistency across metrics and model sizes indicates stable behavior, supporting the view that the dataset itself is saturated rather than that our model is overfitting. We added a new Sec. B, *“Analysis of CCD Performance,”* with Table 9,  in the Supplementary Materials to document these observations.
>
>
> #### Table 9. Comparison of network variants on CCD with additional metrics.
>
> | Model               | AUC (%) | mAP (%) | mTTA (s) | TTA₈₀% | Params (M) |
> |---------------------|---------|---------|----------|--------|------------|
> | Model – smaller (XS) | 85.0    | 99.9    | 4.92     | 4.84   | 13         |
> | Model – small (S)    | 86.9    | 99.9    | 4.95     | 4.86   | 26         |

---

### Official Review · Reviewer_7qEU · 2025-11-01

**Soundness:** 3
**Presentation:** 3
**Contribution:** 3
**Rating:** 6
**Confidence:** 2

**Summary:**

This paper proposes a lightweight dash-cam accident-anticipation model that distills future-aware temporal cues from a frozen MobileCLIP teacher using a windowed RWKV student and masked memory. It addresses the problem of anticipating accidents several seconds earlier than prior work while staying real time on edge devices with only RGB inputs. Experiments show that the porposed method obtains 75.3% mAP and 4.04s mTTA on DAD with 26M params, beating heavier SOTA (in Table 4).

**Strengths:**

- The reviewer found the idea of distilling from an image-only MobileCLIP into a temporal RWKV student for accident anticipation to be interesting.
- Interestingly, temporal-only distillation outperforms spatial-only (74.1% vs 71.2% mAP in Table 2), showing future supervision actually drives anticipation.
- Experiments are comprehensive: Table 1 (RWKV depth), Table 2 (distillation parts), Table 3 (mask ratio), Figure 4 (qualitative attention) and writing is mostly clear and structured.

**Weaknesses:**

- The temporally shifted distillation still does not show a direct comparison to distilling from a video teacher., so it is unclear how much is due to the shift itself.
- Experiments miss an ablation on modern, purely temporal CNN/Transformer baselines that work on DAD/CCD datasets even though Table 4 already cites many frame-level systems.

**Questions:**

- Please respond to the weaknesses above.

---

> ### Author Response · Authors · 2025-11-21
>
> We appreciate the reviewer’s thoughtful comments. Our detailed responses to the raised concerns are provided below.
>
> > - The temporally shifted distillation still does not show a direct comparison to distilling from a video teacher, so it is unclear how much is due to the shift itself.
>
> We thank the reviewer for raising this point. To provide a direct comparison, we added a new Ablation Study (*“Impact of a Video Teacher,”* Table 5) under Sec. 4.3, evaluating distillation from a temporal video teacher (V-JEPA2). Even when the teacher already encodes temporal information, adding our temporal shift still improves anticipation performance (mTTA 4.12 → 4.21 s). This indicates that the benefit arises specifically from the shift mechanism, rather than from the teacher alone, and that the shift is not redundant under temporal supervision.
>
>
> ### Impact of a Video-based Teacher in Temporally Shifted Distillation
> | Video Distillation     | mAP (%) | mTTA (s) |
> |------------------------|---------|----------|
> | VJEPA2 — no shift      | 65.3    | 4.12     |
> | VJEPA2 — shift (1)     | **66.0** | **4.21** |
>
> We also note that V-JEPA2’s lower spatial resolution and different tokenization make spatial distillation less effective than MobileCLIP, resulting in lower mAP overall. This supports that:
>  1. the temporal shift provides a measurable gain even with a video teacher, and
>  2.  strong spatial alignment (as provided by MobileCLIP) remains important.
>
> > - Experiments miss an ablation on modern, purely temporal CNN/Transformer baselines that work on DAD/CCD datasets even though Table 4 already cites many frame-level systems.
>
> Table 8 already includes modern transformer-based video models (VideoSwin, MViTv2, UniFormerV2), which represent the strongest contemporary architectures for DAD/CCD. We did not include purely temporal CNNs (e.g., C3D, I3D) because they are well known to significantly under-perform on accident anticipation tasks. Their coarse temporal pooling and lack of sensitivity to subtle short-range cues (e.g., early deceleration, lane drift, brake-light onset) make them unsuitable for this setting, and recent literature has largely replaced them with transformer-based video models.
>
> This under-performance is expected: accident anticipation relies heavily on fine-grained frame-level cues such as subtle deceleration, lane drifting, or emerging brake lights, whereas purely CNNs use coarse temporal pooling that tends to smooth out these short-range signals, making them less suitable for directly detecting imminent hazards. Adding such weak baselines would not change the comparative conclusions.

---

> > ### Comment · Reviewer_7qEU · 2025-11-26
> > **Response to Authors**
> >
> > Hi Authors,
> >
> > Thank you for your hard work on the rebuttal response.
> >
> > Apologies for overlooking Table 8 (missed it due to the word SOTA in the caption). Can you update the caption to reflect this better? Also, can you cite some works that specifically show "purely temporal CNNs (e.g., C3D, I3D) because they are well known to significantly under-perform on accident anticipation tasks."? Although, the reviewer agrees they are largely replaced them with transformer-based video models, but it would good for the sake of completion.
> >
> > The reviewer may raise the score after some discussions with fellow reviewers.

---

> > > ### Author Response · Authors · 2025-11-28
> > >
> > > Hello Reviewer 7qEU,
> > >
> > > Thank you very much for your constructive feedback. We have updated the caption of Table 8 for clarity, and we have added **additional experiments on purely CNN-based models (I3D, R(2+1)D, TSM)** as suggested. We also revised the related text and included citations indicating that these CNN-based baselines generally under-perform on traffic accident anticipation tasks.
> > >
> > > We hope these updates address your concerns, and we would greatly appreciate your consideration in reflecting them in the score. Thank you again for your thoughtful comments and for your willingness to engage in further discussion.
> > >
> > > #### **Updated Table 8.** Comparison of the highest mAP and corresponding mTTA of priors models on DAD. Best balanced results in terms of model size, accuracy, and early prediction are in bold.
> > > | Method        | Inference Params. (M) | mAP (%) | mTTA (s) |
> > > |---------------|------------------------|---------|----------|
> > > | I3D       | 21                 | 68.0 | 2.99 |
> > > | R(2+1)D   | 31                 | 49.7 | 3.57 |
> > > | TSM       | 43                 | 53.0 | 3.39 |
> > > | GCRNN         | 275                    | 72.2     | 1.33     |
> > > | L-RAI         | 185                    | 51.4     | 3.01     |
> > > | MVITv2        | 51                     | 64.4     | -        |
> > > | DSTA          | 180                    | 72.3     | 1.50     |
> > > | VideoSwin     | 88                     | 65.4     | -        |
> > > | UniFormer V2  | 115                    | 65.2     | -        |
> > > | DAA-GNN       | 183                    | 75.2     | 1.47     |
> > > | MASTTA        | 99                     | 80.8     | 3.32     |
> > > | CCAF-Net      | 191                    | 81.3     | 3.75     |
> > > | **Ours**      | **26**                 | **79.6** | **3.41** |

---

### Official Review · Reviewer_rGQA · 2025-11-02

**Soundness:** 4
**Presentation:** 3
**Contribution:** 3
**Rating:** 6
**Confidence:** 2

**Summary:**

The paper proposes a lightweight framework for accident anticipation using temporally shifted distillation (TSD).
A student model learns to predict future cues by aligning its current features with a teacher’s representation from later frames.
The design combines a RepMixer backbone with an RWKV-based temporal block for efficiency.
Experiments on DAD and CCD show improved early anticipation and real-time performance on edge devices.

**Strengths:**

1.	The idea of learning temporal prediction through shifted distillation is original and intuitively appealing.

2.	The method achieves a good balance between accuracy and efficiency, which makes it suitable for real-world deployment.

3.	The paper is clearly written, and the model architecture and ablation design are well presented.

**Weaknesses:**

1.	The analysis of the temporally shifted distillation is limited.
Table 2 only reports overall performance, so it is difficult to tell whether the model actually learns to anticipate future events or only captures static correlations.
Additional experiments showing temporal behavior, such as feature alignment across time, would strengthen the claim.

2.	The mechanism itself is not deeply analyzed.
It would be useful to evaluate how different time shifts affect learning or whether the model improves its understanding of future frames during training.

3.	Since the teacher model processes only single frames, it is unclear how much of the predictive ability comes from the distillation process compared with the student’s temporal module.
More explanation or comparative evidence would clarify this point.

**Questions:**

Have the authors tried using a video-based teacher that already has temporal understanding? It would be interesting to see whether the proposed shift is still necessary in that setting.

---

> ### Author Response · Authors · 2025-11-22
>
> We appreciate the reviewer’s thoughtful comments. Our detailed responses to the raised concerns are provided below.
>
> > 1. The analysis of the temporally shifted distillation is limited. Table 2 only reports overall performance, so it is difficult to tell whether the model actually learns to anticipate future events or only captures static correlations. Additional experiments showing temporal behavior, such as feature alignment across time, would strengthen the claim.
>
>  We thank the reviewer for this suggestion. To directly verify that temporally shifted distillation teaches future-oriented cues rather than static correlations, we added two new analyses:
>
> 1. **New Temporal-shift Ablation.**
>  We added a new experiment (*“Impact of Temporal Shift in Distillation,”* Sec. 4.3, Table 4), which shows that supervising the student with farther future teacher frames consistently increases mTTA, providing clearer evidence that the model learns stronger anticipatory behavior.
>
>     #### Table 4. Impact of the various temporal shift in temporally shifted distillation.
>     | Temporal Shift            | mAP (%) | mTTA (s) |
>     |---------------------------|---------|----------|
>     | no shift (only spatial)   | 71.2    | 3.79     |
>     | 1                         | **75.3** | 4.04     |
>     | 2                         | 74.7    | 4.06     |
>     | 3                         | 70.7    | **4.13** |
>
> 2. **Feature Alignment Across Time.**
> We further assess the effectiveness of Temporally Shifted Distillation (TSD) by examining the ratio of cosine similarities between two pairs of signals: the Student's features $X_t^S$ aligned with the Teacher's same-time features $X_t^T$, and the Student's features $X_t^S$​ aligned with the Teacher's *future* features $X_{t+1}^T$​. This provides clear evidence that TSD facilitates genuine anticipatory learning rather than mere spatial matching. This analysis is included in a new Section C, *“Analysis of Shifted-Teacher Alignment,”* together with Fig. 5.
>
> Additionally, Table 2 already shows that if the model relied primarily on static alignment, spatial-only supervision would perform better. Instead, temporal-only supervision clearly surpasses spatial-only **(mTTA 3.95 s vs. 3.79 s)**, indicating that the student is learning to predict future representations rather than simply memorizing appearance features
>
> These additional analyses directly support that temporally shifted distillation enhances the model’s temporal reasoning rather than exploiting static correlations.
>
> > 2. The mechanism itself is not deeply analyzed. It would be useful to evaluate how different time shifts affect learning or whether the model improves its understanding of future frames during training.
>
> Point taken. To more deeply analyze the mechanism, we added a dedicated ablation (*“Impact of Temporal Shift in Distillation,”* Sec. 4.3, Table 4). The results show a clear and interpretable trend:
> 1. Shift = 1 → best overall balance (75.3% mAP, 4.04 s mTTA).
> 2. Shift = 2 → slightly lower mAP (74.7%) but **higher mTTA  (4.06 s)**.
> 3. Shift = 3 → further mAP drop (70.7%) but **earliest anticipation mTTA (4.13 s)**.
>
> This indicates that **larger temporal shifts force the student to align with more distant teacher frames**, improving its ability to anticipate earlier but at the cost of weaker spatial fidelity. Conversely, smaller shifts provide the best trade-off between recognition (mAP) and early anticipation (mTTA), consistent with the short predictive horizon characteristic of traffic accident anticipation.
>
> These findings confirm that varying the shift meaningfully changes what the model learns about future frames.

---

> ### Author Response · Authors · 2025-11-22
>
> > 3. Since the teacher model processes only single frames, it is unclear how much of the predictive ability comes from the distillation process compared with the student’s temporal module. More explanation or comparative evidence would clarify this point.
>
> Point taken. The teacher is strictly spatial: it processes each frame independently and provides only appearance features. We then apply a temporal shift so that the teacher’s feature at time $t+1$ supervises the student at $t$. Because the student process the sequence recurrently, it must rely on its Temporal Mixing module—via hidden-state accumulation, learnable decay, and masked updates—to convert past observations into future-predictive features.
>
> To clarify the roles of the student’s temporal module and the shifted supervision, we added a new ablation (*“Impact of Temporal RWKV and Distillation”*) with Table 6. This analysis isolates each component: the temporal module alone offers limited predictive ability, while applying temporally shifted distillation to a spatial-only student already yields a strong gain, showing that the shifted teacher features provide meaningful future-aware cues even without temporal modeling. Crucially, the best performance arises only when both are combined - the temporal module enables recurrence, and the shifted supervision supplies future-frame targets that guide true anticipation.
>
> #### Table 6. Ablation of Temporal RWKV (T-RWKV) and Temporally Shifted Distillation (TSD).
> | Module                      | mAP (%) | mTTA (s) |
> |-----------------------------|---------|----------|
> | T-RWKV + FN                 | 39.4    | 3.97     |
> | S-RWKV + TSD + FN           | 55.6    | 4.00     |
> | T-RWKV + pre-train + FN     | 67.7    | 4.00     |
> | T-RWKV + TSD + FN           | **75.3** | **4.04** |
>
>    (S-RWKV = only spatial recurrence; FN = fine-tuning; pre-train = pre-training without TSD)
>
>
> > 4. Have the authors tried using a video-based teacher that already has temporal understanding? It would be interesting to see whether the proposed shift is still necessary in that setting.
>
> To address this question, we added a new ablation study, ‘"Impact of a Video Teacher"’, and Table 5, where we evaluate distillation from a temporally pretrained video model (V-JEPA2). Even though V-JEPA2 already encodes motion, applying the proposed temporal shift still improves anticipation performance **(mTTA increases from 4.12 s → 4.21 s)**. This demonstrates that the temporal shift provides a complementary benefit and is **not redundant** even in the presence of temporal supervision.
>
> #### Table 5. Impact of a video-based teacher in temporally shifted distillation.
> | Video Distillation     | mAP (%) | mTTA (s) |
> |------------------------|---------|----------|
> | VJEPA2 — no shift      | 65.3    | 4.12     |
> | VJEPA2 — shift (1)     | **66.0** | **4.21** |
>
> We also observe that V-JEPA2 yields lower mAP than MobileCLIP. This is explained by architectural mismatches—V-JEPA2’s lower spatial resolution and different tokenization scheme make spatial alignment more difficult. These results jointly indicate that:
>  1. the temporal shift **remains effective** even when the teacher possesses temporal understanding, and
>  2.  matching the high-quality spatial features with student ones are equally crucial for effective distillation, which MobileCLIP provides, for traffic accident anticipation task.

---

### Author Response · Authors · 2025-12-03

# The Final Summary of Rebuttal Phase

We sincerely thank all reviewers for their constructive and thoughtful feedback, as well as for recognizing the originality, innovation, and comprehensive empirical design of our work. Below is a concise summary of how we addressed every concern raised across **rGQA (R1)**, **7qEU (R2)**, and **bqJU (R3)** during the rebuttal phase.

Across all three reviews, the following points were **consistently highlighted as key strengths of the work**:

   1. **Originality and Innovation of Temporally Shifted Distillation**:  Reviewers agreed that learning temporal prediction via temporally shifted distillation is both novel and intuitively appealing. They highlighted that the approach enables temporal reasoning without relying on video-based pre-trained teachers.

   2. **Effective and Efficient Temporal Modeling:**  The method was repeatedly recognized for achieving a strong balance between accuracy, temporal prediction quality, and computational efficiency—making it suitable for real-world deployment and real-time use under resource constraint devices.

   3. **Comprehensive Experimental Validation:**  All reviewers emphasized the thoroughness of the empirical study, including ablations on:
       - temporal RWKV depth,
       - spatial vs. temporal distillation components,
       - masking ratio,
       - edge device deployment (Jetson Orin Nano),
       - extensive method comparisons, and,
       - qualitative accident-attention analysis.

      These evaluations were noted as well-designed and supportive of the methodological choices.

   4. **Clarity and Presentation Quality:**  Reviewers consistently commended the paper for clear writing, strong organization, and accessible explanations of the architecture and components, along with helpful visual aids.

Across all three reviews, the key requests focused on:

   1. Clarification of architectural choices.
   2. Deeper analysis of the temporally shifted distillation (TSD) mechanism.
   3. Additional ablations and baselines.
   4. Broader validation on large-scale dataset beyond accident anticipation task.


To address these issues comprehensively, we incorporated the following updates:

- ### **Architectural clarifications (R3)**
    We fully clarified our windowing/stitching design, corrected all RWKV-related notation, and expanded our description of the masked-memory mechanism. These updates remove the ambiguity pointed out by R3 and ensure the model formulation and training dynamics are completely transparent.

- ### **New ablations on temporal shift and temporal modeling (R1)**
    We added extensive new experiments, including:
    - *"Impact of Temporal Shift in Distillation"* (Table 4), analyzing different shift magnitudes;
    - *"Impact of Temporal RWKV and Distillation"* (Table 6), isolating the interactions between TSD and the temporal RWKV module;
    - *"C ANALYSIS OF SHIFTED-TEACHER ALIGNMENT"* (Figure 5), showing how student representations align with future teacher features.

    These experiments demonstrate that TSD consistently enhances anticipatory temporal reasoning, directly addressing R1’s concerns.

- ### **New comparisons with a video-based teacher and purely CNN baselines (R2 & R1)**

     We added:
    - distillation experiments using a video-based temporal teacher (V-JEPA2) in *"Impact of Video Teacher"* (Table 5), showing that the proposed temporal shift yields improvements even when the teacher already encodes temporal structure;
   - purely CNN baselines (I3D, R(2+1)D, TSM) integrated into the updated Table 8.

   These additions strengthen the position of TSD against both stronger teachers and classical video models.

- ### **Additional CCD analysis and smaller model variants (R3)**
   In *"B ANALYSIS OF CCD PERFORMANCE"* (Table 9), we provide additional metrics AUC, TTA@80, and XS-model results.

   This analysis confirms that the near-ceiling mAP arises from CCD dataset saturation, not overfitting, directly resolving R3’s concerns.

- ### **Larger-scale evaluation (R3)**
    We added results on Kinetics-400, demonstrating that our model remains effective beyond domain-specific accident datasets (Official Comment for R3, point 4).

    At the same time, we emphasize that the core motivation of TSD is efficient pre-training for small-to-medium-scale domains without requiring a video-based teacher, which the new results strongly support.


## **Closing Statement**
All concerns raised by R1, R2, and R3 have been fully addressed. The newly added analyses, ablations, and clarifications significantly strengthen the technical soundness and empirical validity of the paper. We believe the revised submission is substantially improved thanks to the reviewers’ insightful feedback, and we hope this will be reflected in the AC’s final evaluation.

We appreciate your careful consideration.

---

### Meta-Review · Area_Chair_EqaM · 2026-01-06

**Summary:**

All three reviewers acknowledged the novelty, practicality, and strong empirical results of the paper, particularly praising the temporally shifted distillation (TSD) framework and its suitability for real-time, edge-device deployment. Initial concerns centered on: (1) insufficient analysis of whether TSD truly enables anticipation learning; (2) lack of comparison with video-based teachers and classical CNN baselines; (3) architectural ambiguities; (4) potential overfitting on the saturated CCD dataset; and (5) generalizability beyond accident anticipation.

**Reviewer Concerns:**

The authors thoroughly addressed every concern in the rebuttal:
- Authors provided new ablations showing that larger temporal shifts increase mTTA, confirming predictive learning.
- Authors introduced the Temporal Alignment Ratio (TAR), proving student features align more strongly with future teacher features.
- Authors added experiments with a video-based teacher (V-JEPA2), showing TSD still improves mTTA.
- Authors included classical baselines, demonstrating their good performance on this task.
- Authors clarified the RWKV design, and confirmed the masking strategy incurs negligible cost and is used only in pre-training.
- Authors confirmed performance stems from the CCD dataset's saturation, not overfitting.
- Authors reported results on Kinetics-400, showing the architecture generalizes beyond the target domain.
- All concerns were resolved with evidence-backed updates that significantly strengthened the paper’s rigor and completeness.

**Reviewer Scores:**

- Reviewer rGQA (score 6)
- Reviewer 7qEU (score 6) acknowledged the rebuttal and indicated willingness to raise the score.
- Reviewer bqJU (score 4) may revise upward to 6 after all technical issues were clarified.

---

### Decision · Program_Chairs · 2026-01-26

Accept (Poster)